# Gastroprotection against Rat Ulcers by *Nephthea* Sterol Derivative

**DOI:** 10.3390/biom11081247

**Published:** 2021-08-21

**Authors:** Tarik A. Mohamed, Abdelsamed I. Elshamy, Mahmoud A. A. Ibrahim, Mohamed A. M. Atia, Rania F. Ahmed, Sherin K. Ali, Karam A. Mahdy, Shifaa O. Alshammari, Ahmed M. Al-Abd, Mahmoud F. Moustafa, Abdel Razik H. Farrag, Mohamed-Elamir F. Hegazy

**Affiliations:** 1National Research Centre, Chemistry of Medicinal Plants Department, 33 El−Bohouth St., Dokki, Giza 12622, Egypt; ta.mourad@nrc.sci.eg (T.A.M.); sheryali57@yahoo.com (S.K.A.); me.fathy@nrc.sci.eg (M.-E.F.H.); 2National Research Centre, Chemistry of Natural Compounds Department, Dokki, Giza 12622, Egypt; ai.el-shamy@nrc.sci.eg (A.I.E.); rfawzi@hotmail.com (R.F.A.); 3Computational Chemistry Laboratory, Chemistry Department, Faculty of Science, Minia University, Minia 61519, Egypt; m.ibrahim@compchem.net; 4Molecular Genetics and Genome Mapping Laboratory, Genome Mapping Department, Agricultural Genetic Engineering Research Institute (AGERI), Agricultural Research Center (ARC), Giza 12619, Egypt; matia@ageri.sci.eg; 5National Research Centre, Medical Biochemistry Department, 33 El Bohouth St., Dokki, Giza 12622, Egypt; karammahdy64@gmail.com; 6Department of Biology, College of Science, University of Hafr Al Batin, Hafar Al Batin 39524, Saudi Arabia; Dr.shifaa@uhb.edu.sa; 7Department of Pharmaceutical Sciences, College of Pharmacy & Thumbay Research Institute for Precision Medicine, Gulf Medical University, Ajman 4184, United Arab Emirates; 8Pharmacology Department, Medical Division, National Research Centre, Cairo 12622, Egypt; 9Department of Biology, College of Science, King Khalid University, Abha 9004, Saudi Arabia; hamdony@yahoo.com; 10Department of Botany & Microbiology, Faculty of Science, South Valley University, Qena 83523, Egypt; 11National Research Centre, Pathology Department, 33 El Bohouth St., Dokki, Giza 12622, Egypt; ar.hussein@nrc.sci.eg

**Keywords:** *Nephthea* species, soft corals, anti-ulcer activity, molecular docking, reactome analysis

## Abstract

Different species belonging to the genus *Nephthea* (Acyonaceae) are a rich resource for bioactive secondary metabolites. The literature reveals that the gastroprotective effects of marine secondary metabolites have not been comprehensively studied in vivo. Hence, the present investigation aimed to examine and determine the anti-ulcer activity of 4*α*,24-dimethyl-5*α*-cholest-8*β*,18-dihydroxy,22*E*-en-3*β*-ol (**ST-1**) isolated from samples of a *Nephthea* species. This in vivo study was supported by in silico molecular docking and protein–protein interaction techniques. Oral administration of **ST-1** reduced rat stomach ulcers with a concurrent increase in gastric mucosa. Molecular docking calculations against the H^+^/K^+^-ATPase transporter showed a higher binding affinity of **ST-1**, with a docking score value of −9.9 kcal/mol and a p*K*_i_ value of 59.7 nM, compared to ranitidine (a commercial proton pump inhibitor, which gave values of −6.2 kcal/mol and 27.9 µM, respectively). The combined PEA-reactome analysis results revealed promising evidence of **ST-1** potency as an anti-ulcer compound through significant modulation of the gene set controlling the PI3K signaling pathway, which subsequently plays a crucial role in signaling regarding epithelialization and tissue regeneration, tissue repairing and tissue remodeling. These results indicate a probable protective role for **ST-1** against ethanol-induced gastric ulcers.

## 1. Introduction

Gastric ulcers are an erosion of the stomach lining caused by disruptions of the gastric mucosal defense and/or repair systems [1]. While such ulcers are one of the most prevalent gastrointestinal disorders, available treatments usually focus on reducing gastric acid production and re-enforcing gastric mucosal defenses. Gastric H^+^/K^+^-ATPases underlie the establishment of the highly acidic environment in the stomach (pH ≈ 1), which is necessary to promote the indispensable digestion of proteins in food [2]. The enzyme further plays an essential repression of this proton/potassium pump, as it limits acids produced by the stomach. Therefore, such repressors can be categorized as anti-ulcer drugs [3]. While peptic ulcer-healing drugs for gastric ulcers act as inhibitors of proton pumps and enhance the integrity of the mucosal barrier [4], some lack selectivity and/or produce side effects [5,6].

Galectin-3 (Gal3) is a biomarker highly represented by activating macrophages and several cell types constitutively, such as gastrointestinal epithelial cells [7,8,9]. In patients with gastric cancer, serum levels of galectin-3 are significantly increased, in contrast with both patients with benign disease and healthy control subjects. Serum galectin-3 level may serve as a therapeutic target for this disease [10]. A second biomarker monitored in this study is tumour necrosis factor alpha (TNF alpha), which acts as an inflammatory cytokine generated by macrophages/monocytes throughout acute inflammation and is liable for a wide range of signaling functions within cells, driving either apoptosis or necrosis. This protein was also found to play an integral role in resistance to infection and cancers [11].

Currently, there is a resurging interest in plant and marine-derived products being used as natural medicines. Indeed, a considerable number of medicinal plants and dietary nutrients have been shown to possess gastroprotective effects [12,13,14,15,16]. Natural compound research studies and coral reef communities are regarded as the best reservoirs of potential novel chemical entities that may benefit from their biological properties [17,18,19,20,21]. Soft corals belonging to *Nephthea* genus (Acyonaceae) are a wide-spreading species in the Red Sea and are a rich resource of steroids, terpenoids and ceramides [22]. Different species of this genus have been described as promising sources of natural compounds with diverse pharmacological and biological activities, comprising anti-inflammatory, anticancer [18,22,23,24], and anti-microbial potentialities [25]. A previous investigation of *Nephthea* species revealed the identification of several biologically active and drug development leads [2,19,22,23], such as cytotoxic potentiality towards several tumor cell lines [22,26,27,28,29]. Recently, pathway enrichment analysis (PEA) and reactome analysis were developed to helps scientists and researchers gain biological-wide insights through gene lists generated from omics-scale experiments. These methods recognize the biological pathways, discover any unexpected functional relationships, and finally deliver the results in interactive networks with highly enhanced diagrams of drug-target interactions [25]. The molecular docking technique is a widely used in silico tool for predicting the binding affinities and modes of inhibitors/drug candidates with biological targets.

Since the biological importance of marine secondary metabolites is well-recognized, we isolated secondary metabolite (4*α*,24-dimethyl-5*α*-cholest-8*β*,18-dihydroxy,22*E*-en-3*β*-ol, **ST-1**) from *Nephthea* species to continue our search for bioactive products for the treatment of gastric disorders utilizing natural resources. Herein, the gastroprotective activity of the isolated steroid, 4*α*,24-dimethyl-5*α*-cholest-8*β*,18-dihydroxy,22*E*-en-3*β*-ol (Figure 1), from Red Sea *Nephthea* sp. was evaluated against ethanol-induced rats based upon biochemical and histopathological analyses compared with ranitidine as the reference drug. Moreover, the molecular docking technique was utilized to predict the binding mode and evaluate the potentiality of the isolated steroid against the H^+^/K^+^-ATPase transporter.

## 2. Materials and Methods

### 2.1. Soft Coral Material, Extraction, Isolation, Purification and NMR Spectroscopy

The Egyptian Red Sea soft coral *Nephthea* sp. (voucher specimen: 399RS-9100-X19) was collected and authenticated by Dr. Montaser A. Alhammady, National Institute of Oceanography and Fisheries, Marine Biological Station, Hurghada, Egypt, from the Hurghada coast in May 2019. Four kilograms of the frozen soft coral were cut, extracted via 6 L of a mixture of 1:1 (MeOH–CH_2_Cl_2_), filtered, and dried under vacuum-afforded dark brown gum (167.8 g) using a rotary evaporator. The extract was further fractionated over silica gel column chromatography (6 × 120 cm) using *n*-hexane (100%), a step gradient of *n*-hexane–CH_2_Cl_2_, CH_2_Cl_2_ (100%), and CH_2_Cl_2_–MeOH until CH_2_Cl_2_–MeOH (1:1) afforded seven main fractions (NP-I to NP-VII). Fraction NP-4 (3.1 g) was subjected to silica gel column chromatography and generated three sub-fractions (NP-4A-C). The sub-fraction NP-4B (673.4 mg) was eluted by CHCl_3_-MeOH (1:1) over a glass column packed with Sephadex LH-20 (3 × 120 cm) to afford compound **ST-1** (181.7 mg). Compound **ST-1** was analyzed via (i) NMR spectroscopic analysis (600 MHz Bruker NMR spectrometer, USA) in CD_3_OD and (ii) mass spectroscopy (JEOL JMS-600 instrument (Tokyo, Japan) (Appendix A). The chemical structure of **ST-1** was constructed by comparison of its NMR with previously published data (Appendix A) [22].

### 2.2. Experimental Animals, Ethical Statement, Ulcer Induction, and Grouping

This study was performed on healthy female Wistar rats of 12–16 weeks (150–180 g) that were obtained from the animal lab at the National Research Centre, Dokki, Cairo, Egypt. Rats were held in polypropylene cages under standardized rearing conditions (room temperature: 22 ± 2 °C, 55 ± 5% humidity with 12 h dark/light cycles). Rodents were fed a pellet-based diet and allowed free access to water. Rats were divided into five groups with six animals each as follows: group 1: control rats; group 2: ethanol ulcerated group; group 3: ulcerated rats pretreated with 30 mg/kg ranitidine (as a reference drug); group 4: ulcerated rats pretreated with 50 mg/kg of **ST-1**; and group 5: ulcerated rats pretreated with 100 mg/kg of **ST-1** (dissolved in distilled water with a few drops of DMSO). After 2 hrs of all treatments with ranitidine (group 3) and compound **ST-1** (group 4 and 5), the rats were orally administrated 1 mL EtOH to induce ulceration. This analysis was conducted by several studies [30,31,32]. For the ulceration induction, the fasted rats (*n* = 6) were orally administrated with 1 mL EtOH (99.9% purity) [33]. After 4 h, sacrificing of all rats occurred under anesthesia by diethyl ether. The stomachs were harvested, washed in saline solution and dry-blotted. Subsequently, the stomachs were fixed in formalin saline (10%) for histological investigation [34,35].

The animal experiments were conducted according to the international regulations of usage and welfare of laboratory animals and were approved by the Ethics Committee of the National Research Centre, Cairo, Egypt (Approval No: 18/204) [32].

### 2.3. Galactin-3 and TNF-α Determination

Serum galectin-3 and TNF-α levels were determined by enzyme-linked immunosorbent assay technique using kits purchased from Sun Red Biotechnology (Shanghai, China). The operational processes were measured in accordance with the kit’s instructions.

#### 2.3.1. Histopathology Study

At the end of the experiment, a small piece of the stomach from each rat was excised and fixed in formalin, dehydrated in ascending grade of ethanol and embedded in paraffin wax. Sections 5 µm thick were cut in a microtome and mounted on glass slides using standard techniques. After staining the tissues with hematoxylin-eosin stains, the slides were viewed under a light microscope equipped for photography.

#### 2.3.2. Gastric Mucosal Glycoprotein Evaluation

To examine gastric mucosal glycoproteins, stomach sections (5 µm) were stained with periodic acid–Schiff (PAS) to observe changes in glycoproteins [36].

#### 2.3.3. Statistical Analysis

All results were expressed as means ± SE. The data were calculated using SPSS 19.0 (SPSS Inc., Chicago, IL, USA). The statistical significance of differences for each parameter between groups was evaluated by one-way ANOVA, followed by the LSD test. The significance level was set at *p* < 0.05.

### 2.4. Molecular Docking

The PDB file for the H^+^/K^+^-ATPase α chain was downloaded from the SwissProt database (https://swissmodel.expasy.org, accessed on 12 January 2021) (ID: P20648). A previous published protocol for molecular docking of natural metabolites to target proteins was followed [37,38,39,40]. Docking parameters were set to 250 runs and 25,000,000 energy evaluations for each cycle. Docking was performed by Autodock 4.2.6 software using the Genetic Algorithm [41]. Docking was performed three times independently to calculate mean values and standard deviations of the lowest binding energies and predicted inhibition constants (p*k*_i_). The representation and graphical analyses were performed using the VMD software.

### 2.5. Protein–Protein Interaction

The online web-based tools of SwissTargetPrediction (http://www.swisstargetprediction.ch, accessed on 21 May 2021) were applied to predict the biological targets for SP1 as a gastric ulcer inhibitor. The DisGeNET online database (https://www.disgenet.org, accessed on 21 May 2021) was utilized to collect the available database for Gastric ulcer diseases. A Venn diagram was designed using the InteractiVenn online tool [42]. A protein–protein interaction (PPI) network was generated using a STRING functional database for the top predicted targets [43]. Cytoscape 3.8.2 was employed to investigate target-function relations based on the network topology [44]. Furthermore, to explore all probable target-function relations for the top 20 targeted genes based on their biological network mining, pathway enrichment analysis was performed using Cytoscape 3.8.2. Finally, the ReactomeFIViz tool was utilized for modeling and visualization of the **ST-1**-target interactions [45].

## 3. Results

### 3.1. Biochemical Results

Rodents pretreated with **ST-1** showed a significant reduction in galactin-3 and TNF-α disease progression biomarkers in comparison with an ethanol-treated group. Ranitidine used as a positive control also exhibited a significant reduction in both galactin-3 and TNF-α biomarkers, as revealed in Table 1. In contrast, serum galactin-3 and TNF-α increased in rats treated with ethanol alone. However, these acute studies have provided information regarding only acute phase responses and instant adaptation of the stomach to toxic insults.

### 3.2. Histopathological Results

Examination of sections of the stomach of rats from the normal control group shows mucosa with integral surface mucosal epithelium and no lesions which have developed (Figure 2A). In case of the ulcerated group, microscopic investigation showed severe disruption of the surface epithelium. Necrotic lesions in the mucosa layer associated with hemorrhagic erosion were seen (Figure 2B). Histological study of stomach mucosa of sections of the ulcerated group treated with 50 mg/kg of **ST-1** showed mucosa appeared more similar to the control group (Figure 2C). In some rats, this group showed mild mucosal surface erosion and no edema (Figure 2D). On the other hand, mucosa of the ulcerated group treated with 100 mg/kg of **ST-1** showed intact surface mucosal epithelium and no visual lesions (Figure 2E). In case of the EtOH-induced, ranitidine-treated group, mild erosion of mucosa was observed (Figure 2F).

### 3.3. Histochemical Results

Periodic acid–Schiff reagent was used to stain polysaccharide material. Histochemical assessment of untreated stomach sections showed that the gastric mucosa was mostly localized to the epithelium covering the stomach mucosa. Extensive stain is recognized in the apical zones of these cells, such that a thick coat of magenta color continues along the stomach epithelium luminal surface (Figure 3A). In rodents that received a single oral dose of ethanol to induce ulcers, both epithelial and mucous neck cells’ deteriorated surface was relatively devoid of stainable material (Figure 3B). In the ulcerated group that received 50 mg/kg of **ST-1**, the outer half of the fundic mucosa showed dense staining as compared with that of the group which received ethanol only (Figure 3C). In some tissue, the mucosa exhibited weak stainable material (Figure 3D). In the ulcerated group that received 100 mg/kg of **ST-1**, heterogeneous staining was confronted where the deteriorated surface of epithelial and mucous neck cells was relatively devoid of stainable material, while the outer half of the fundic mucosa was heavily stained (Figure 3E). No change was observed in the stainability of the polysaccharides in the fundic mucosa of ulcerated tissue administrated with a single oral dose of ranitidine. The polysaccharide contents of mucosal epithelium were almost similar to that of the control rats (Figure 3F).

### 3.4. In Silico Inhibitory Effect of ST-1 on H^+^/K^+^-ATPase

Anti-ulcer drugs have the ability to prevent gastric acidification as the main factor of ulcer formation via gastric H^+^/K^+^ ATPase. Accordingly, we investigated the binding mode and affinity of **ST-1** with H^+^/K^+^ ATPase; ranitidine was used as a positive control [33]. **ST-1** showed a higher affinity to the proton pump than ranitidine (−9.86 vs. −6.21 kcal/mol, respectively) and exhibited a higher estimated inhibition constant (p*k*_i_ values 59.72 nM vs. 27.92 µM, respectively). **ST-1** showed interaction with GLN133, ASN143, ASN144, LEU147, VAL344, ALA345, VAL347, GLU801, THR819, ILE822, and ASP826 amino acid residues (Figure 4).

### 3.5. Molecular Target Prediction and Network Analysis

**ST****-1** protein targets were initially predicted and categorized with the aid of SwissTargetPrediction (Figure 5). Then, with the help of DisGeNET online tools, one hundred and seventeen genes were recognized in terms of gastric ulcer diseases (C0038358). The Venn diagram comparison results showed that shared genes for SP1 included CYP2C19, CYP2C9, PTGS1, KDR, MET, MAPK1, and NOS2 (Figure 6). CYP2C19 and CYP2C9 are the most polymorphic enzymes related to CYP2C genes in humans, and they metabolize many important clinical drugs, including anti-ulcer drugs. Increased nitric oxide synthase (NOS2) can lead to large amounts of NO secretion and induce severe damage to many kinds of tissues. Inhibition of NOS2 leads to a decrease in its level in the gastric mucosa [46]. **ST-1-**predicted gene targets were also analyzed via a STRING PPI network and visualized by Cytoscape 3.8.0. The top 20 scored genes for SP1 included CYP2C19, CYP2C9 and NOS2 (Table 2).

### 3.6. Pathway Enrichment Analysis (PEA)

For better and deeper mining/dissection of **ST-1** target-function interactions, a Voronoi tree map based on Boolean network modeling and PEA analysis was achieved. The Voronoi tree map was constructed to visualize the top targeted pathway influenced by the top 20 gene targets in response to **ST-1** (Figure 7). Furthermore, a reactome graphical representation was built on the top pathway affected in response to **ST-1** (Figure 8). Notably, pathways involved in diseases of signal transduction by growth factor receptors and second messengers, as well as disease and immune system pathways, were found to be the most significant pathways targeted by **ST-1**, with a false discovery rate (FDR) of <0.00001% (Table 3).

Interestingly, under the “diseases of signal transduction by growth factor receptors and second messengers” pathway, it was found that the PI3K signaling pathway was the most enriched pathway induced by **ST-1** treatment among the human biological pathways. Dissection of the PEA analysis combined with reactome mining emphasized that a set of ten genes (PIK3CA, PIK3CB, PIK3CD, ESR1, KIT, MTOR, HSP90AA1, MAPK1, KDR, and MDM2) were significantly modulated as biological targets to **ST-1** as a potent anti-ulcer drug. Moreover, the PEA-reactome outcomes disclosed that these ten genes were found to significantly interact with 13 other biological interactors/genes, including P29353, P46531, P48729, Q00987, P31749, P00533, P62993, P27986, P19174, Q07817, Q96B36, and P42336.

## 4. Discussion

Gastric ulceration is one of the most common gastrointestinal disorders in clinical practice. While, in many cases, the etiology of the ulcer is still unexplained, it is generally affirmed that aggressive factors of both pepsin and acid can maintain mucosal integrity throughout endogenous defensive mechanisms. To recover a healthy balance, therapeutics alone or together with plant-derived medications have been used [47]. This study evaluated both biochemically and microscopically the gastroprotective effects of 4*α*,24-dimethyl-5*α*-cholest-8*β*,18-dihydroxy,22*E*-en-3*β*-ol (Figure 1) isolated from *Nephthea* sp. on ethanol-induced gastric damage in vivo. The results were compared with ranitidine as a reference anti-ulcerogenic drug.

Gastric ulcers, when ethanol-induced, act as widespread ulcerogenic agents. Ethanol is processed in the body to release superoxide anion and hydroperoxy free radicals. The oxygen-resultant free radicals were found to be involved in the stomach ulcer mechanism [40]. Several mechanisms are implicated in ulcer origin in many models. Experimental proof has shown that antioxidants can promote gastric wall protection and safeguard tissue from oxidative damage [46]. Additionally, gastric acid excretions exhibited a role in the formation of gastric ulcers. Furthermore, substances which can modulate gastric acid secretion, such as proton pump suppressors and histamine H2 receptor antagonists, are deemed to accelerate the healing process of gastric lesions or repress the creation of mucosal injury [47].

Histologically, there were no stomach lesions for the control group in contrast to the ethanol-induced treatment, which showed severe ulceration and hemorrhage. Upon ranitidine pre-treatment, the mucosal epithelium had less hemorrhage and erosion. Lesions of gastric mucosa were also reduced with combined EtOH and **ST-1** treatment, which may be attributed to reduced oxidative damage and leukotriene activity.

Inflammation is regarded as the essential marker for stomach ulceration. A feature of pathogenesis of peptic ulcers is the imbalance between offensive factors, such as gastric acid, and protective factors, including inflammatory cytokines. Ethanol has been known to upregulate pro-inflammatory markers and to downregulate anti-inflammatory biological facets [48]. The examined levels of TNF-α in gastric tissue were found to be highly expressed in the ethanol-induced ulcer model while in the 4*α*,24-dimethyl-5*α*-cholest-8*β*,18-dihydroxy,22*E*-en-3*β*-ol isolated from *Nephthea* sp. pretreated rats, gastric tissue exhibited a significant decrease in the expression of TNF-α.

PI3K encompasses a family of lipid kinases that are classified based on their capability to activate inositol phospholipids. PI_3_K-dependent activation of AKT was strongly confirmed to affect the activity of numerous downstream biological pathways involved in apoptosis, cell proliferation, cellular survival, senescence, and angiogenesis [49,50]. Additionally, it is associated with cancer progression. PI3K/AKT signals epithelialization [51], tissue regeneration and repairing [52], as well as tissue remodeling [53]. Deregulation of PI_3_K/AKT signaling, in contrast, can compromise wound healing [54]. Remarkably, PI_3_K/AKT signaling was reported to switch on or off numerous downstream regulating proteins involving glycogen synthase kinase 3 (GSK3) and mammalian target of rapamycin (mTOR). The phosphorylation of GSK3 and mTOR by PI_3_K/AKT signaling stimulate a broad range of biological activities, including growth, proliferation, and survival [55].

In this study, galactine 3 showed a significant increase in the ethanol-induced ulcer group; on the other hand, in rats pretreated with **ST-1**, levels of Gal3 were reduced. Gal3 has been shown to have a variety of pro-inflammatory and anti-microbial functions (Hsu, et al. 2000; Beatty, et al., 2002), including macrophage activation, which plays a role in survival and phagocytosis of macrophages/neutrophils (6–8), as well as neutrophil extravasation [56,57].

## 5. Conclusions

The present study affords the first perceptions into the gastroprotective potential of 4*α*,24-dimethyl-5*α*-cholest-8*β*,18-dihydroxy,22*E*-en-3*β*-ol **(ST-1)** in ethanol-induced gastric ulcers in rats. **ST-1** protects rodent gastric mucosa from ethanol-induced ulcers in vivo.

From the bioinformatical interpretation, molecular docking calculations against H^+^/K^+^-ATPase transporter showed a higher binding affinity of **ST-1** with a docking score value of −9.9 kcal/mol and a p*K*_i_ value of 59.7 nM. Additionally, the molecular target prediction and network analysis showed that the PI3K signaling pathway was the highest enriched pathway induced by **ST-1** treatment among the human biological pathways. The combination between PEA analysis and the reactome mining toolbox (which gives insights about biological pathways based on massive experimentally validated datasets merged with deep in silico analysis) showed significantly modulated genes, particularly genes involved in the PI3K signaling pathway, as biological targets of **ST-1** as a potent anti-ulcer. These results indicate a probable protective role for **ST-1** against ethanol-induced gastric ulcers, and the anti-ulcerogenic effect of **ST-1** will require additional investigations to determine its mechanism of action.

## Figures and Tables

**Figure 1 biomolecules-11-01247-f001:**
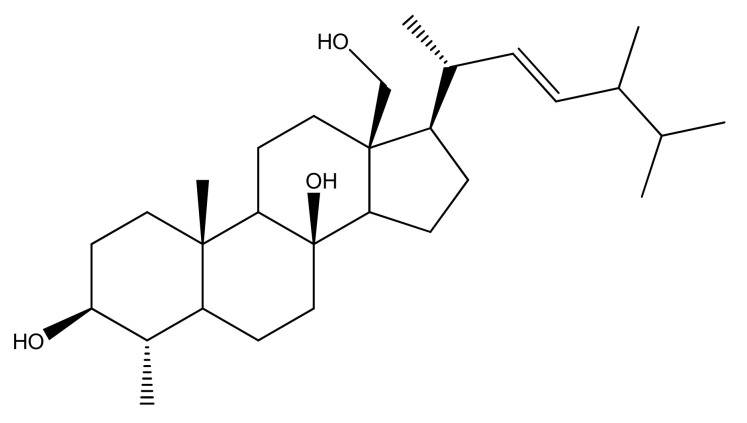
Chemical structure of 4*α*,24-dimethyl-5*α*-cholest-8*β*,18-dihydroxy,22*E*-en-3*β*-ol (**ST-1**).

**Figure 2 biomolecules-11-01247-f002:**
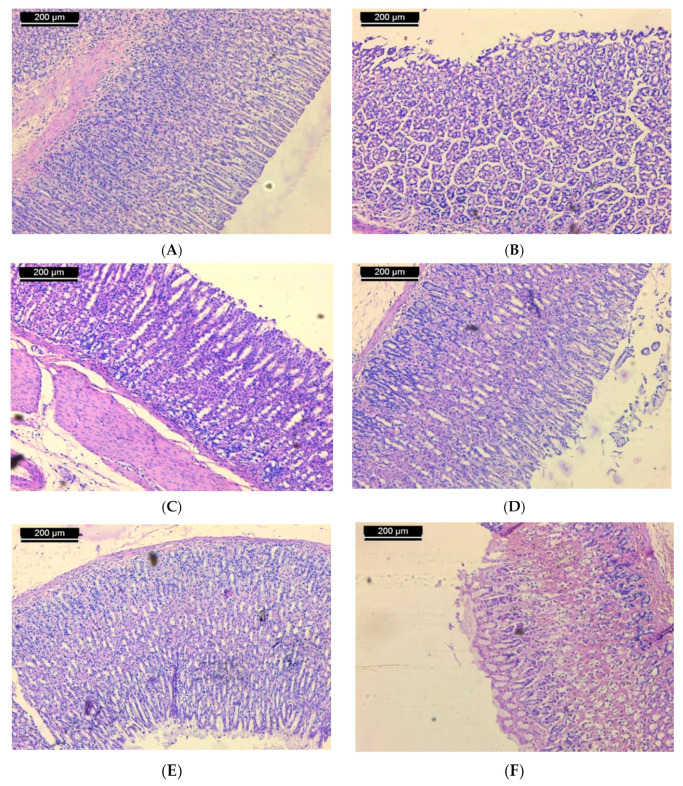
Micrograph of sections of the stomach of rats from (**A**): normal control shows mucosa with intact surface mucosal epithelium and no lesion appeared; (**B**): ulcerated group shows severe disruption of the surface epithelium and necrotic lesions mucosa with hemorrhagic erosion, (**C**): the group treated with 50 mg/kg of **ST-1** shows mucosa appeared more like the control (**D**): the group treated with 50 mg/kg of **ST-1** shows mild mucosal surface erosion and no edema; (**E**): the group treated with 100 mg/kg of **ST-1** shows intact surface mucosal epithelium and no lesions which have appeared; (**F**): the group treated with ranitidine, showing mild erosion of mucosa (H & E stain, Scale Bar; 200 µm).

**Figure 3 biomolecules-11-01247-f003:**
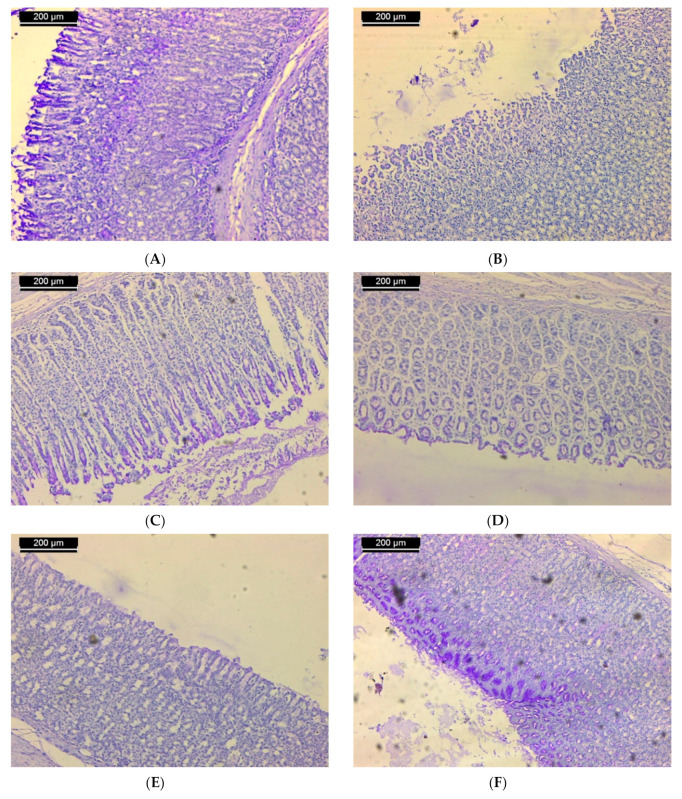
Sections of the stomachs of rats from: (**A**) the normal control group showed the polysaccharide material in the gastric mucosa. These materials are localized in the mucosa epithelium; (**B**) the ulcerated group showed a lack of stainable material; (**C**) the group treated with 50 mg/kg of **ST-1** showed dense staining as compared with the group which received ethanol only; (**D**) the group treated with 50 mg/kg of **ST-1** exhibit weak stainable material; (**E**) the group treated with 100 mg/kg of **ST-1** showed heterogeneous staining, such that the degenerated surface epithelial cells and mucous neck cells were almost devoid of stainable material, while the outer half of the fundic mucosa was densely stained; (**F**) the group treated with ranitidine showed that the polysaccharide contents of the mucosal epithelium were almost similar to that of the control rats (PAS stain; scale bar, 200 µm).

**Figure 4 biomolecules-11-01247-f004:**
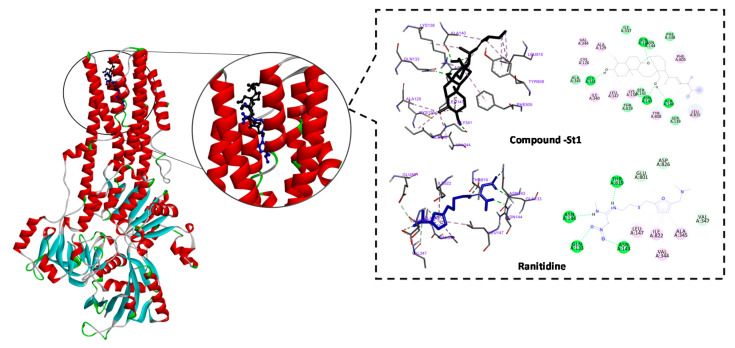
2D and 3D representations of the predicted binding modes, as well as the docking scores, of **ST-1**.

**Figure 5 biomolecules-11-01247-f005:**
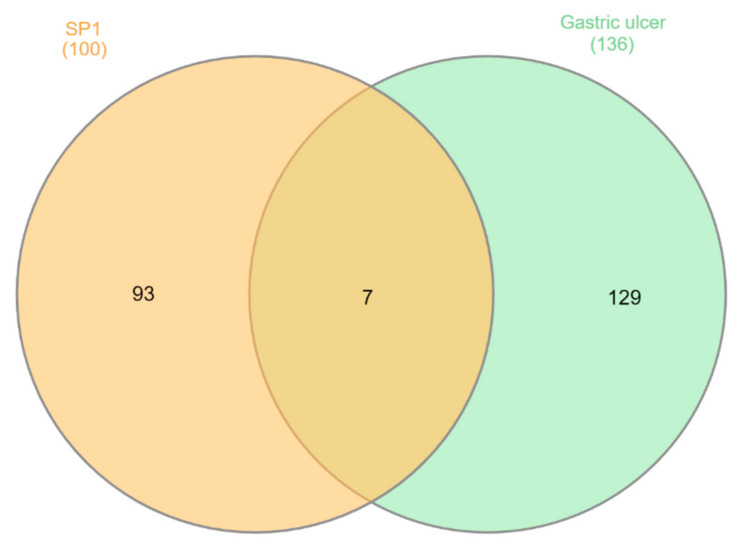
Venn diagram analysis for **ST-1** and gastric ulcer disease (C0038358) genes.

**Figure 6 biomolecules-11-01247-f006:**
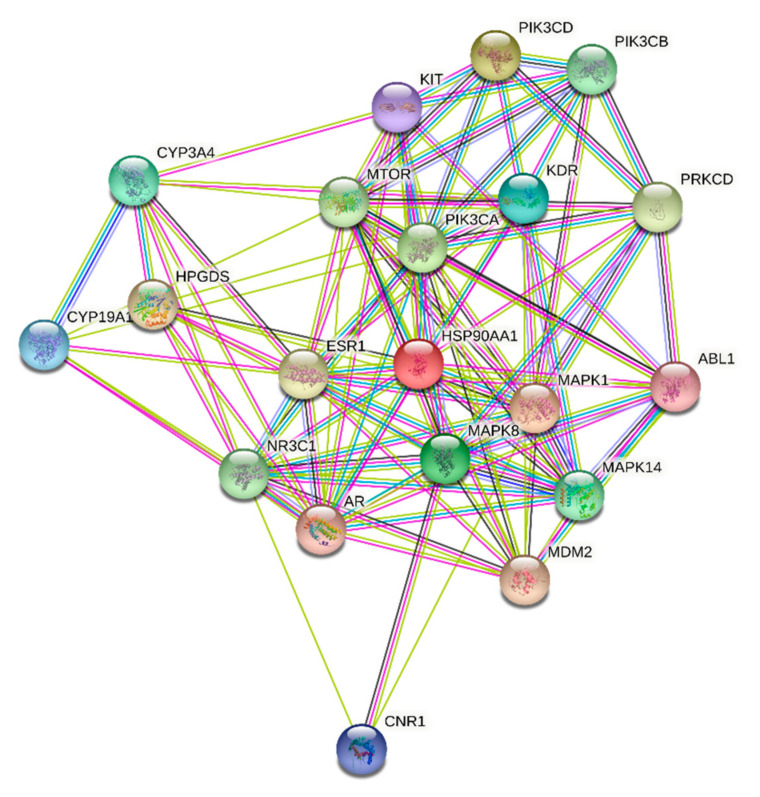
STRING PPI network for the top 20 targets identified by network analyzer for **ST-1** as a potent gastric H+/K+-ATPase inhibitor.

**Figure 7 biomolecules-11-01247-f007:**
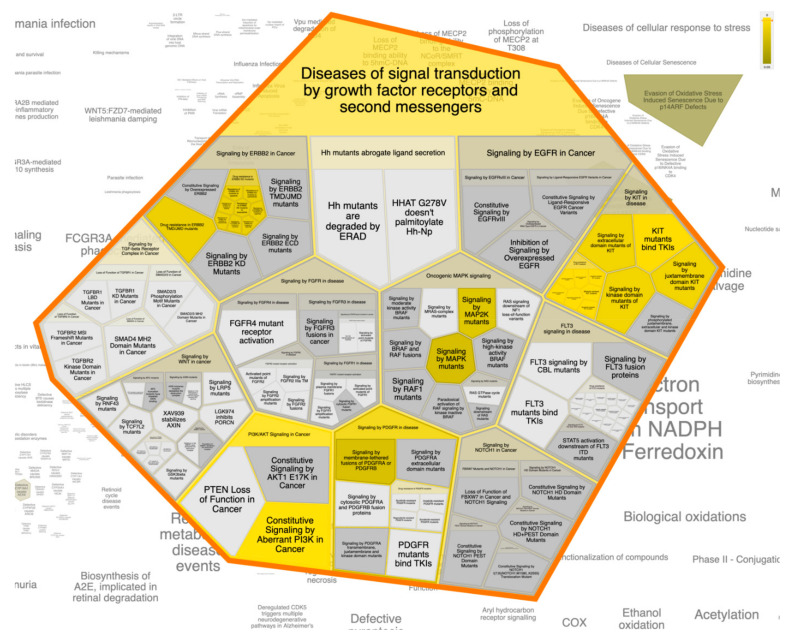
The Voronoi treemap of the top pathway (diseases of signal transduction by growth factor receptors and second messengers) influenced by the top 20 gene targets in response to **ST-1**.

**Figure 8 biomolecules-11-01247-f008:**
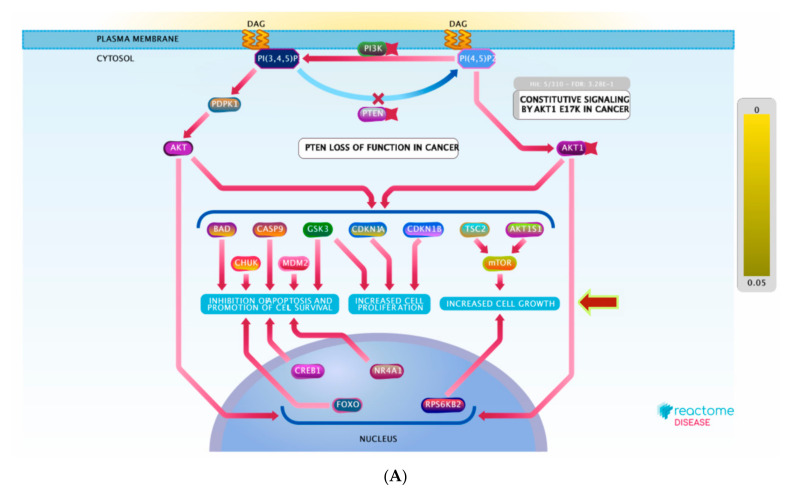
(**A**) Graphic representation of the diseases of signal transduction by growth factor receptors and (**B**) second messengers reactome pathway, influenced as a response to **ST-1** in the human genome.

**Table 1 biomolecules-11-01247-t001:** Effects of oral administration of **ST-1** on blood Galactin-3 (ng/mL) and TNF-α (Pg/mL) levels in ethanol-induced gastric ulcer in rats.

Parameters	Galactin-3 (ng/mL)		TNF-α Pg/ml	
Groups	Mean ± S.E	% Change	Mean ± S.E	% Change
Control	1.64 ± 0.11	0	29.82 ± 0.56	0
Ethanol (1 mL)	13.99 ± 0.14 ^a^	+753 ^a^	209.37 ± 2.91 ^a^	+602 ^a^
Ranitidine	2.25 ± 0.09 ^ab^	−83.9 ^b^	42.58± 0.84 ^ab^	−79.7 ^b^
**ST-1** (50 mg)	12.68 ± 0.32 ^ab^	−9.4 ^b^	193.44 ± 2.55 ^ab^	−7.6 ^b^
**ST-1** (100 mg)	9.52 ± 0.32 ^ab^	−32 ^b^	168.81 ± 6.32 ^ab^	−19.4 ^b^

Data presented as means ± standard error (no. 6); ^a^ Significantly different at the *p* < 0.05 and % of change as compared with control group; ^b^ Significantly different at the *p* < 0.05 and % of change as compared with ulcer group.

**Table 2 biomolecules-11-01247-t002:** Network topological analysis for the predicted targets for **ST-1**.

Name	Betweenness Centrality ^a^	Closeness Centrality ^b^	Degree ^c^
MAPK1	0.12998495	0.585987261	34
MTOR	0.05082444	0.531791908	31
PIK3CA	0.04725076	0.51396648	30
HSP90AA1	0.057040046	0.519774011	28
ESR1	0.03944331	0.525714286	28
MAPK8	0.096876023	0.541176471	26
AR	0.033861461	0.508287293	25
CYP3A4	0.066526563	0.508287293	23
KDR	0.03944376	0.50273224	22
NR3C1	0.059041577	0.528735632	21
MDM2	0.015376587	0.476683938	20
MAPK14	0.01274391	0.484210526	18
HPGDS	0.03677825	0.489361702	18
PRKCD	0.015162524	0.446601942	17
CNR1	0.082987497	0.469387755	17
CYP19A1	0.031696473	0.471794872	17
PIK3CB	0.004766941	0.433962264	16
ABL1	0.006113177	0.444444444	15
PIK3CD	0.013300606	0.427906977	15
KIT	0.006066222	0.46	14

^a^ Betweenness quantifies the number of times a node acts as a bridge along the shortest path between two other nodes. ^b^ Closeness represents the highly-connected network, indicating the influencers in a single cluster. ^c^ Degree is the simplest measure of node connectivity.

**Table 3 biomolecules-11-01247-t003:** Top 20 pathways for ST1 targets resulting from the pathway enrichment analysis (PEA).

Pathway Name	# Entities Found	# Interactors Found	Entities *p*-Value	# Reactions Found	Submitted Entities Hit Interactor
Diseases of signal transduction by growth factor receptors and second messengers	18	22	0.62086893	269	MTOR; KIT; PIK3CA; MAPK8; NR3C1; MAPK1; MAPK8; MAPK14; PIK3CD; HSP90AA1; MAPK14; PRKCD; KDR;ABL1; ESR1; PIK3CD; MDM2; ESR1; AR; PIK3CB; PIK3CA; KDR
Disease	12	17	0.0033075	197	MTOR; KIT;MAPK8; NR3C1; MAPK1; PIK3CD; HSP90AA1; PRKCD; ABL1; ESR1; PIK3CD; MDM2; AR; ESR1; PIK3CB; KDR;PIK3CA
Immune system	12	18	0.96689721	150	MTOR; KIT; MAPK8; MAPK1; NR3C1; PIK3CD; MAPK14; HSP90AA1; PRKCD; ABL1; PIK3CD; ESR1; MDM2; AR; PIK3CB; PIK3CA; HPGDS; KDR
Nuclear receptor transcription pathway	11	2	3.75 × 10^−14^	2	ESR1; MDM2
Intracellular signaling by second messengers	11	7	0.00100386	17	MTOR; HSP90AA1; KIT; MAPK8; NR3C1; MDM2; ESR1
PIP3 activates AKT signaling	10	6	0.00133626	13	MTOR; HSP90AA1; MAPK8; NR3C1; MDM2; ESR1
Cellular responses to stress	10	13	0.09103042	83	MTOR; MAPK8; NR3C1; MAPK1; MAPK14; HSP90AA1; MDM2; ABL1; MDM2; ESR1; AR; KDR; PIK3CA
Cellular responses to external stimuli	10	13	0.10148968	83	MTOR; MAPK8; NR3C1; MAPK1; MAPK14; HSP90AA1; MDM2; ABL1; MDM2; ESR1; AR; KDR; PIK3CA
Signaling by receptor tyrosine kinases	10	20	0.3190745	227	MTOR; KIT;MAPK8; NR3C1; MAPK1; CNR1; PIK3CD; MAPK14; HSP90AA1; PRKCD; KDR;ABL1; ESR1; PIK3CD; ESR1; MDM2; PIK3CB; KDR;PIK3CA; HPGDS
Cytokine signaling in the immune system	10	17	0.63708745	79	KIT; MAPK8; MAPK1; NR3C1; PIK3CD; MAPK14; HSP90AA1; PRKCD; ABL1; PIK3CD; MDM2; ESR1; AR; PIK3CB; PIK3CA; HPGDS; KDR
PI3K/AKT signaling in cancer	9	5	4.44 × 10^−6^	18	MTOR; HSP90AA1; NR3C1; MDM2; ESR1
PI5P, PP2A and IER3 regulate PI3K/AKT signaling	8	3	2.68 × 10^−6^	4	MTOR; NR3C1; MDM2
Negative regulation of the PI3K/AKT network	8	3	5.31 × 10^−6^	4	MTOR; NR3C1; MDM2
Axon guidance	8	12	0.31029712	31	KIT; HSP90AA1; MAPK8; MAPK14; PRKCD; ABL1; ESR1; NR3C1; MAPK1; MDM2; ESR1; KDR
Nervous system development	8	12	0.39349828	32	KIT; HSP90AA1; MAPK8; MAPK14; PRKCD; ABL1; ESR1; NR3C1; MAPK1; ESR1; MDM2; KDR
Signaling by interleukins	8	12	0.4667549	51	HSP90AA1; KIT; MAPK8; PRKCD; ABL1; ESR1; MDM2; MAPK14; PIK3CB; PIK3CA; KDR; HPGDS
Innate immune system	8	11	0.73193686	66	HSP90AA1; KIT; MAPK8; PRKCD; ABL1; MAPK1; ESR1; MDM2; AR; MAPK14; PIK3CA
Developmental biology	8	15	0.92886656	60	KIT; MAPK8; NR3C1; MAPK1; NR3C1; HSP90AA1; ESR1; MAPK14; PRKCD; ABL1; ESR1; ESR1; MDM2; AR; KDR
Metabolism	8	12	0.99897572	71	MTOR; NR3C1; HSP90AA1; PIK3CA; MDM2; ESR1; NR3C1; MAPK1; AR; MDM2; ESR1; MAPK14
Constitutive signaling by aberrant PI3K in cancer	7	0	1.40 × 10^−6^	2	MTOR; KIT;PIK3CA; MAPK8; NR3C1; MAPK1; MAPK8; MAPK14; PIK3CD; HSP90AA1; MAPK14; PRKCD; KDR;ABL1; ESR1; PIK3CD; MDM2; ESR1; AR; PIK3CB; PIK3CA; KDR

## Data Availability

The data presented in this study are available in the article/Appendix A.

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
