# Peer review of "Gastroprotection against Rat Ulcers by Nephthea Sterol Derivative"

_biomolecules, 2021, doi:10.3390/biom11081247_

Round 1

Reviewer 1 Report

The manuscript presents a experimental and theoretical study about the effect of a natural compound from an algae species on protection agains gastric ulcer. 

The manuscript is well-written and presented, however there are some aspects that need to be imporved:

Materials and Methods, section 2.1: the chromatography equipment must be described. Given de amounts of purified compound I think the equipment is a preparative one? This shpuld be clarified.

The use of molecular simulation is not fundamented enough. This should be clarified in the introduction.

Table 2: explain the meaning of the parameters.

Fig. 7 and 8: improve quality/resolution, enlarge typography, they are not readable.

Author Response

Dear Reviewer,

Kindly find attached the point-by-point response to your valuable comments

Reviewer 2 Report

With the original research article biomolecules-1297250, Hegazy and colleagues deliver in vivo experimental data on the potential protective effects of a sterol derivative against ethanol-induced gastric ulcer. The experimental design has been properly conceived and while the results have some scientific soundness the current study certainly merits further data to corroborate the potential gastroprotective effects of the compound. A series of issues that hamper the acceptance of the current version of the manuscript are stressed out below:

  1. The manuscript requires extensive English language editing, authors being advised to have it revised by a native speaker or an English editing service. Indeed, several sentences are unclear as in:

Lines 45-47: “Gastric H+/K+-ATPases are causing the underlie the establishment of the highly acidic environment in the stomach (pH ≈ 1) to promote food consumption and the indispensable digestion of proteins in food.”. The enzyme It further plays an essential…”.

Lines 50-52: “While many peptic ulcer healing drugs gastric ulcer aim to act as inhibitors of proton pumps and enhance the integrity of the mucosal barrier [4], some lack selectivity such pharmaceuticals can be non-specific and/or produce side effects [5,6].”.

Additional examples evidencing an uncareful writing style can be found throughout the whole manuscript.

  1. Authors deliver very personal points of view that lack scientific validation as in:

Lines 65-67: “Natural compounds research studies and coral reef communities are regarded as the best reservoirs of potential novel chemical entities that may benefit from their biological properties [17-19].”

Is there any scientific rationale (chemoecological for instance) suggesting that coral reef communities are the best reservoirs of novel chemical entities?

  1. While the sterol derivative has been previously reported by the authors (DOI: 10.1080/14786419.2015.1055266), it has been isolated from a different collection of Nephthea sp. samples. As such, and in order to fully validate de structural identity of the compound, authors are requested to include at least the 1D-NMR data (i.e. 1H and 13C-NMR; for example as supplementary material).

  1. Authors should further clarify the potential gastroprotective effects of 4α,24-dimethyl-5α-cholest-8β,18-dihydroxy, 22E-en-3β-ol. While claiming that galactin-3 levels have been reduced after exposure to the compound (Lines 156-157), a significant increase is observed at 50 mg Kg-1 in comparison with the control (i.e., ethanol-induced gastric ulcer animal models) (Table 1).

  1. The study lacks experimental evidence on the observed effects, namely the characterization of the mechanisms underling the potential gastroprotective properties that are suggested by the histochemical and morphological analyses (Sections 3.2 and 3.3.). Indeed, while the molecular docking study, as well as the target prediction and network analysis, might provide clues on the molecular targets of the compound, such results are merely indicative (and oftentimes speculative). Authors should have considered such analyses to drive and further deepen their experimental design, including the in vitro evaluation of the effects towards iNOS2 (Line 229) or H+/K+ATPase, both quite attainable (i.e., cheap and easy to perform).

  1. Finally, authors did not generate any experimental data on the antioxidant or “anti-secretory” effects of the sterol derivative in order to conclude that such effects “appear to play some role in the pharmacological activity of this natural product(Lines 330-331).

Author Response

Dear Reviewer,

Please find attached a point-by-point response to your valuable comments.

Reviewer 3 Report

The manuscript Gastroprotection against rat ulcers by Nephthea sterol derivative submitted by the authors fits the journal’s scope.  The authors present the results of the anti-ucer activity testing of an isolated compound from the Nephthea species. The evaluation included an in vivo experiment and in silico studies, and two potential mechanisms for the anti-ulcer activity were formulated. Although the in silico assessment is described in detail, the authors should add more information and clarifications regarding the in vivo experimental design.

The section 2.2. Experimental animals and grouping should be updated considerably. The references are missing;

The rats ‘bred is not specify;

the ulcer induction protocol is not presented;  

What international regulations were consulted?

Why were selected only 6 animals/group (when 8 is the minimum accepted number)?

How were the doses selected?

ST-1 was administered in single-dose?

How was the isolated compound solubilized for oral administration?

How were collected the biological samples and how long after the oral administration and the ulcer induction?

Why the authors did not performed a determination of free HCl and an assay for peptic activity?

2.3.2. Histopathology study - Please indicate the references.

In order to formulate this conclusion, other in vivo experimental evidences are needed (at least free HCl and peptic activity).

Author Response

Dear Reviewer,

Please find attached a point-by-point response to your valuable comments

Round 2

Reviewer 2 Report

Despite addressing some issues raised by the reviewers, it is my personal opinion that the revised version of the manuscript biomolecules-1297250 delivered by Hegazy and colleagues still lacks a major revision. Alternatively, and considering that the results are preliminary and that the gastroprotective effects can be considered as weak, the limited scientific novelty might be sufficient to have the study published as a communication rather than as a full original research article.

Firstly, the manuscript still requires an extensive English language (and style) editing as, for example, evidenced in:

Line 32: Revise to “…isolated from samples of a Nephthea sp.”.

Lines 57-58: Revise to “…the serum levels of galectin-3 are significantly increased in patients with gastric cancer, in contrast with both…”.

Line 67: Revise to “…to possess gastroprotective effects…”.

It also worth to emphasize that the selected key words “biochemical” or “histological” are unspecific i.e., too broad, and not representative of the main scientific outcomes of the current study.

Authors now made available 1D and 2D NMR data that appears to corroborate the proposed structure. However, the main limitation of the manuscript deals with the very weak gastroprotective activity of the steroid derivative, namely on the interference with galactin-3 and TNF-α levels. Indeed, oral administration of the constituent at 50 mg/Kg was found to increase galactin-3 levels (Table 1) i.e. further contributing to the ethanol-induced toxicity. Additionally, and while significant, the reduction of galactin-3 levels after 100 mg/Kg administration led to a weak effect (4-fold lower than that observed with ranitidine). Similarly, reduction of TNF-α levels upon administration of the steroid derivative are weak, nearly negligible in comparison with ranitidine (Table 1), thus reducing the scientific relevance of the study.

In response to my previous comment, calling for further experimental evidence on the mechanisms underlying the weak gastroprotective effects, authors solely included additional data dealing with the experimental design underlying the in vivo studies ?! It should be noted that molecular docking analysis are merely indicative and act as a guide to select and tune the experimental design on targets to be selected for in vitro (or in vivo) testing. As such, I still consider that it would be worth to investigate the effects of the steroidal constituent upon iNOS2 and H+/K+ATPase.       

Author Response

Response to Reviewer 2 Comments

Point 1: Line 32: Revise to “…isolated from samples of a Nephthea sp.”.

Reply:  As recommend by the reviewer, the sentence was revised.

Point 2: Lines 57-58: Revise to “…the serum levels of galectin-3 are significantly increased in patients with gastric cancer, in contrast with both…”.

Reply:  As recommend by the reviewer, the sentence was revised.

Point 3: Line 67: Revise to “…to possess gastroprotective effects…”.

Reply: As recommend by the reviewer, the sentence was revised and the keywords “biochemical” or histological” were replaced by Reactome analysis. updated.

Point 4: In response to my previous comment, calling for further experimental evidence on the mechanisms underlying the weak gastroprotective effects, authors solely included additional data dealing with the experimental design underlying the in vivo studies ?! It should be noted that molecular docking analysis are merely indicative and act as a guide to select and tune the experimental design on targets to be selected for in vitro (or in vivo) testing. As such, I still consider that it would be worth to investigate the effects of the steroidal constituent upon iNOS2 and H+/K+ATPase.       

Reply: The authors thank the reviewer for his/her comment. However, we do not have the required facilities to perform the requested experiments at the current time. On the other hand, we used a smart and powerful approach utilizing a combination between PEA analysis and Reactome-mining toolbox. The utilized approach gives insights about biological pathways and their biological roles based on massive experimentally validated Omics-datasets such as transcriptomic, proteomic, …etc., merged with deep in silico analysis. As presented in the manuscript, this approach revealed a significant modulation of gene-set involved in the PI3K Signaling pathway, which subsequently plays a crucial role in signals epithelialization and tissue regeneration, tissue repairing, and tissue remodeling, as a result of treatment with ST-1 as a potent anti-ulcer.

Reviewer 3 Report

The authors clarified the majority of raised issues, especially those regarding the design and the protocol. However, before publication, some minor corrections are needed. Please see them below:

Point 2. The rats ‘bred is not specified

Please specify if Wistar rats were used

Point 4. What international regulations were consulted?

Reviewer comment (Round 2): Please indicate (add the references) the international regulation guidelines

Point 5. Why were selected only 6 animals/group (when 8 is the minimum accepted
number)?

Response: There are many published articled included 5-8 rats; Also we have several published articles in highly ranked journal with thi

Reviewer comment (Round 2): Please add in the manuscript a justification for using groups of 6 rats.

Point 6. How were the doses selected?

Response: The doses of compound 1 were selected according to the acute toxicity test.

Reviewer comment (Round 2): Please add a paragraph in the manuscript.

Point 8. How was the isolated compound solubilized for oral administration?
Response: This compound was soluble in water with few drops of DMSO

Reviewer comment (Round 2): Please add this to section 2.2.  

Point 10. Why the authors did not perform a determination of free HCl and an assay for
peptic activity?
Response: With the respecting to the reviewer comment, we have not the facilities for this

Reviewer comment (Round 2): Please use these determinations in future works!

Author Response

Response to Reviewer 3 Comments

Thanks to the respected reviewer for his comments to improve our manuscript.

Point 2. The rats ‘bred is not specified

Reply:   This study was performed on healthy female Wistar rats. The reviewer’s comment was considered in the revised manuscript.

Point 4. What international regulations were consulted?

Reviewer comment (Round 2): Please indicate (add the references) the international regulation guidelines

Reply:  The reviewer’s comment was considered in the revised manuscript, and the related reference was added.

Point 5. Why were selected only 6 animals/group (when 8 is the minimum accepted
number)?

Reviewer comment (Round 2): Please add in the manuscript a justification for using groups of 6 rats.

Reply:  The related references were added

Point 6. How were the doses selected?

Reviewer comment (Round 2): Please add a paragraph in the manuscript.

Reply: A paragraph was added to the revised manuscript.

Point 8. How was the isolated compound solubilized for oral administration?

Reviewer comment (Round 2): Please add this to section 2.2.  

Reply:   Sentence was added